# Preventing chronic malnutrition in children under 2 years in rural Angola (MuCCUA trial): protocol for the economic evaluation of a three-arm community cluster randomised controlled trial

Rocio Martin-Cañavate [1,2] Estefania Custodio [1,3] Elena Trigo,[4]
María Romay-Barja [1,3] Zaida Herrador,[5,6] Isabel Aguado,[4] Ferran Ramirez,[4]
Lourdes Maria Faria,[7] Ana Silva-Gerardo,[8] Jose Carlos Lima,[8] Eva Iráizoz,[9]
Tayná Marques,[4] Antonio Vargas,[9] Amador Gomez,[9] Chloe Puett,[10] Israel Molina[4]

**Correspondence to**
Dr Estefania Custodio;
ecustodio@isciii.es

## ABSTRACT

**Introduction** Chronic malnutrition is a serious problem in southern Angola with a prevalence of 49.9% and 37.2% in the provinces of Huila and Cunene, respectively. The MuCCUA (Mother and Child Chronic Undernutrition in Angola) trial is a community-based randomised controlled trial (c-RCT) which aims to evaluate the effectiveness of a nutrition supplementation plus standard of care intervention and a cash transfer plus standard of care intervention in preventing stunting, and to compare them with a standard of care alone intervention in southern Angola. This protocol describes the planned economic evaluation associated with the c-RCT.

**Methods and analysis** We will conduct a cost-efficiency and cost-effectiveness analysis nested within the MuCCUA trial with a societal perspective, measuring programme, provider, participant and household costs. We will collect programme costs prospectively using a combined calculation method including quantitative and qualitative data. Financial costs will be estimated by applying activity-based costing methods to accounting records using time allocation sheets. We will estimate costs not included in accounting records by the ingredients approach, and indirect costs incurred by beneficiaries through interviews, household surveys and focus group discussions. Cost-efficiency will be estimated as cost per output achieved by combining activity-specific cost data with routine data on programme outputs. Cost-effectiveness will be assessed as cost per stunting case prevented. We will calculate incremental cost-effectiveness ratios comparing the additional cost per improved outcome of the different intervention arms and the standard of care. We will perform sensitivity analyses to assess robustness of results.

**Ethics and dissemination** This economic evaluation will provide useful information to the Angolan Government and other policymakers on the most cost-effective intervention to prevent stunting in this and other comparable contexts. The protocol was approved by the República de Angola

## STRENGTHS AND LIMITATIONS OF THIS STUDY

⇒ This protocol will contribute to the growing evidence regarding cost-effective and cost-efficient multisectoral interventions to prevent chronic malnutrition.
⇒ The study design draws on a comprehensive and broad cost analysis based on diverse data sources such as trial data, community-level surveys, staff interviews, interviews and focus group discussions with participants, and detailed programme expenditures.
⇒ The economic evaluation design will allow societal, participant, household and provider costs to be disaggregated.
⇒ Community-based trials pose methodological challenges in terms of data collection and participant follow-up especially in these rural and remote settings of southern Angola.
⇒ The economic evaluation will be conducted in the context of a research project, and costs could be influenced by the research component in ways that cannot be seen or corrected for.

Ministério da Saúde Comité de Ética (27C.E/MINSA. INIS/2022). The findings of this study will be disseminated within academia and the wider policy sphere.
**Trial registration number** ClinicalTrials.gov Registry (NCT05571280).

## BACKGROUND
### Burden of child undernutrition
Stunting or chronic malnutrition is the impairment of growth and development that children experience due to malnutrition, repeated infections and inadequate psychosocial stimulation.[1] Stunting in early life, particularly in the first 1000 days, from conception

through the first 2 years of life, has adverse physical and functional consequences in the child including higher risk of morbidity and mortality, lower educational attainment and cognition, lower adult earnings, loss of productivity, and when accompanied by excessive weight gain in childhood, increased risk of nutrition-related chronic diseases in adulthood.[2] Stunting affects 149.2 million children under 5 years of age. The number of children with stunting is declining in all regions except Africa.[3]

In Angola, the level of chronic malnutrition is high, with 38% of children being stunted and 5% of children being acutely malnourished nationally (Demographic and Health Survey 2016–2017).[3] According to UNICEF, 3.8 million people in Angola face food and nutrition insecurity due to climate shocks.[4] Recurrent drought, reduced agricultural and livestock production, crop failures, high food prices and the proliferation of infectious diseases all contribute to compromising maternal and infant nutrition and health.[5 6] The southwestern provinces of Angola have been most severely affected. In the provinces of Cunene and Huila, the latest Integrated Food Security Phase Classification analysis revealed that over 1.3 million people are estimated to be experiencing high levels of acute food insecurity with more than 50% of their total population in crisis or emergency levels between April 2021 and March 2022.[6] The situation has deteriorated during the lean season as food reserves were depleted, drought conditions continue to be exacerbated by erratic below-average rainfall and the humanitarian response has proven inadequate to address the most urgent needs.[7] Further, a Standardised Monitoring and Assessment of Relief and Transitions Survey conducted in 2019 in selected municipalities of Huila and Cunene reported a prevalence of global acute malnutrition of 11% in both provinces, and a stunting prevalence of 49.9% and 37.2% in the provinces of Huila and Cunene, respectively.[8]

Due to the severity of the situation, there is a need to improve the nutritional status of this population. In relation to stunting, there is growing evidence that nutrition-specific interventions such as small-quantity lipid-based nutrient supplementation (SQ-LNS) have benefits on child growth.[9] The positive effects of SQ-LNS on growth was validated by a meta-analysis that provided evidence of this effect at reducing stunting or wasting in a variety of contexts, recommending to policymakers its inclusion in intervention packages.[10] There is also evidence that nutrition-sensitive interventions, for example, cash transfers, could improve nutrition and health outcomes and accelerate reductions in stunting.[11 12] The Crescer Project aims to address stunting in southern Angola by implementing a multisectoral approach with both a nutrition-specific and a nutrition-sensitive intervention.

## Mother and Child Chronic Undernutrition in Angola trial

The Crescer Project is an operational research programme on the prevention of chronic malnutrition in the southern Angolan provinces of Huila and Cunene. It is embedded in the FRESAN (Fortalecimento da Resiliência e da Segurança Alimentar e Nutricional em Angola) Programme, funded by the European Union (European Development Funds) to strengthen resilience and food and nutrition security in southern Angola. The Crescer Project implements the MuCCUA (Mother and Child Chronic Undernutrition in Angola) trial, a three-arm community-based cluster randomised controlled trial of non-inferiority which aims to evaluate the effectiveness and cost-effectiveness of a nutrition supplementation plus standard of care intervention and a cash transfer plus standard of care intervention compared with a standard of care alone intervention in preventing stunting in children at 24 months in Huila and Cunene. Pregnant women and their newborn children are target recipients of either arm: the comparator arm receives standard of care alone, the second arm receives standard of care plus nutrition supplementation with SQ-LNS and a family food ration, and the third arm receives standard of care plus unconditional cash transfers. Community and Health Development Agents (ADECOS) implement the interventions following Angolan Government protocols.[13] The primary outcome is prevalence of stunting in children. The MuCCUA trial is described in detail elsewhere.[14] The purpose of this protocol paper is to fully describe the methodology for the economic evaluation of the trial.

## Economic evaluations of multisectoral interventions to prevent child stunting

Delivering multisectoral interventions combining nutrition-specific and nutrition-sensitive interventions during the first 1000 days of life is key to prevent chronic malnutrition and to improve early child growth.[15] Reduction of stunting requires improvements in food and nutrition security, education, water, sanitation and hygiene interventions, health, the status of women and poverty reduction.[2] Maternal and child interventions such as nutritional supplementation with SQ-LNS,[10 16–18] cash transfers,[12 19 20] vitamin A supplementation and behaviour change communication have shown promising results in preventing chronic malnutrition.[21 22] However, evidence on the effectiveness of combining them in multisectoral programmes is scarce.[23]

In order to advocate for further investment on these multisectoral programmes, there is a need to provide knowledge not just on their effectiveness but also on their costs,[24] as funding agencies are increasingly requesting implementers to include economic evaluations in their proposals.

The body of economic evidence supporting nutrition programmes is growing,[15] but it is still limited especially for low-income and middle-income countries (LMICs)[25 26] and for multisectoral nutrition interventions.[15]

Multisectoral programmes including health, nutrition, cash and behaviour change communication interventions pose challenges for economic evaluations due to their heterogeneity. First, studies with interventions delivered in different sectors can be difficult to compare due to differences in the outcomes and costs included.[27–30]

Second, the perspective chosen for the economic evaluation poses another challenge for interpretation and comparison of findings between studies because of the different sets of costs included.[25 31 32] Third, economic evaluation objectives differ greatly between studies as some aim to prioritise policies and funds, and different studies may aim to inform different decision-makers.[28]

There are published studies on the costs of complex interventions including nutrition supplementation[33 34] or cash transfer interventions,[11 35] but to the best of our knowledge, there are no studies comparing the costs of these two interventions within the same project, as this study does.

Finally, a recent review on cost analysis of nutrition interventions on LMICs highlighted the need to standardise methods and reporting in economic evaluations in order to facilitate interpretation and provide a means for comparing costs and cost-effectiveness of interventions.[36] The initiative SEEMS-Nutrition (Strengthening Economic Evaluation for Multisectoral Strategies for Nutrition) has developed a common approach guide to measure the costs and benefits of multisectoral nutrition strategies[37] that we have followed.

The proposed economic evaluation of the MuCCUA trial aims to address the challenges mentioned above by assessing the cost-efficiency and cost-effectiveness of the interventions following the SEEMS guide. The perspective used will be societal including all costs of the interventions regardless of who incurs them.

Thus, we hope to contribute to the scarce literature on the cost-efficiency and the cost-effectiveness of implementing multisectoral nutrition interventions including both nutrition-specific and nutrition-sensitive interventions, with a standardised methodology in an LMIC real-world setting located in southern Angola.

## Aim and objectives
The aim of the MuCCUA trial economic evaluation is to use a societal perspective to estimate the costs, and evaluate the cost-efficiency and cost-effectiveness of three interventions (standard of care alone, standard of care plus nutritional supplements and standard of care plus cash transfers) in reducing stunting in children under 2 years of age from poor and very poor households in the southern provinces of Cunene and Huila in Angola.

The specific objectives of the economic evaluation are the following:
1. To estimate the costs of setting up and implementing each of the MuCCUA interventions.
2. To calculate costs incurred by the intervention participants and their families as a result of the interventions.
3. To estimate the cost-efficiency of each of the three interventions as cost per woman, child and household reached.
4. To estimate the cost-effectiveness as cost per case of stunting averted by comparing incremental costs and incremental changes in the outcomes of the interventions, compared with the standard of care.

In addition to the above objectives, heterogeneity analysis of the MuCCUA interventions will be assessed to evaluate how the costs and consequences of the interventions are distributed among different subgroups of the study population.

## METHODS
### Study design
A cost-efficiency and cost-effectiveness analysis of the MuCCUA trial, a community-based cluster randomised trial that compares three interventions to prevent children's stunting taking place between October 2022 and February 2025. We will estimate the total and incremental costs of the interventions prospectively from a societal perspective, measuring programme, provider, participants and household costs.

### Patient and public involvement
Community leaders (locally named 'sobas') and the community were involved at the early stages of the research with discussions about the outcomes and the interventions. Then, ADECOS were selected by their own communities only after fulfilling certain technical and capacity criteria. Sobas, communities and ADECOS were involved in the cluster randomisation. Methods of recruitment were informed by discussions with community activity assistants and ADECOS through a focus group session. All women from selected randomised communities included were identified and informed of the study by ADECOS.

Pregnant women and their families were not involved in setting the research question or the outcome measures. The project is promoting understanding and dissemination of key messages around the interventions to motivate community involvement during and beyond the study. A feedback/suggestions mechanism for participants and communities has been implemented as well.

At the end of the study, results will be shared and discussed with participants, their families and communities through dedicated workshops. Results and all the knowledge generated should promote change at community, regional and national level in policies and practices.

### Study setting and population
The MuCCUA trial is set in four communes of Huila and Cunene, two provinces of southern Angola. These provinces are predominantly rural with a combined total population of around 4.4 million (Huila population: 3 185 244 and Cunene population: 1 271 638).[38] There are province-specific differences related to livelihoods. Huila has a higher agricultural productivity and lower food insecurity risk as compared with Cunene, mostly livestock activity, where the semiarid conditions and frequent droughts occasionally result in significant livestock loss, a key livelihood asset. Overall, the population is highly dependent on staple foods (primarily maize, millet and sorghum) and there is poor access to safe drinking water,

hygiene and sanitation which may differ by commune also, due to the communes' different geographical features. The socioeconomic status of clusters and households is expected to be homogeneous, as one of the clusters' inclusion criteria was to experience extreme poverty. Different ethnic groups are present in the clusters but we do not expect diverse access to interventions according to ethnic origin as none of this groups are nomad. However, access to healthcare and other essential services differs in relation to the distance from the communities to the main settlements of each of the communes. There are remote communities where long distance, lack of transport and lack of trained health professionals and medicines make access to basic services inadequate.

The study participants are pregnant women confirmed by pregnancy test and over 16 years of age, and their newborn children. The target population of the interventions is the household in which the pregnant woman and their children live. In the case of having more than one pregnant woman within the household, all women who accept to participate will be included. We consider households to be the person or group of persons, with or without kinship relationships, who have been habitually living under the same roof for at least 6 months or less but with the intention of staying in the residence for the next 6 months and share food and/or other vital needs such as water, cooking and eating utensils and hygiene products. Informed cluster-level consent was sought from village leaders, sobas, to participate in the trial before randomisation. After acceptance to participate in the study through the informed consent, mothers and their children are followed up from the time of pregnancy at the time of recruitment until the child is 24 months old.

### Trial design

In the MuCCUA trial, a total of 36 purposively selected clusters (the selection of which is described below), with an estimated population of around 55 000, were randomised to the three trial arms with six clusters per arm in Huila province, and six clusters per arm in Cunene province. Two municipalities (admin 2) were selected per province (admin 1) with one commune (admin 3) per municipality (total two communes per province). In the province of Huila, the commune of Libongue (municipality of Chicomba) and the commune of Jamba Sede (municipality of Jamba) were selected. In the province of Cunene, the commune of Otchinjau (municipality of Cahama) and the commune of Mupa-Mukolongodjo (municipality of Cuvelai) were selected. The cluster unit is typically a village or neighbourhood and any nearby hamlets within each commune or, in the case of large neighbourhoods, a distinct part of it that is similar in population size to the unit. The randomisation, which took place in September 2022, was stratified by cluster.

The inclusion criteria for clusters were being considered as one of the municipalities prioritised by the FRESAN Programme and having a multidimensional poverty level 4 or 5 according to the Instituto Nacional de Estatística classification,[39] not currently having or planned to have in operation other interventions offering monetary or nutritional transfers, acceptance by the municipal and traditional authorities, accessible in a 4×4 vehicle and having a reference health post with reference health personnel.

Recruitment and baseline survey started in October 2022 and will continue until March 2023. Implementation started at recruitment and is planned to end in November 2025. A total of 1440 pregnant women and their newborns, 480 in each arm, are enrolled. Primary efficacy analyses will be performed when the participant's children have completed a total of 24 months of follow-up. A full description of the intervention design within the MuCCUA trial is described in detail elsewhere.[14] A brief description of the interventions is presented in the following sections.

### MuCCUA interventions

#### Standard of care arm

This is the comparator arm. The communities allocated to this arm receive as intervention the standard of care alone, as described below.

The standard of care intervention comprises the recruitment of ADECOS and the implementation of a set of activities described in the National Development Plan of Angola[40] and in the Multisectoral Strategic Plan on Nutrition,[41] aligned with the WHO maternal and infant healthcare recommendations, following the National Policy of ADECOS. This intervention will be used as a comparator as it fulfils the criteria of acceptability, feasibility, relevance and uniformity required to be a good comparator in this type of studies.[42] The standard of care activities can be grouped into two blocks:

1. Health promotion activities: including community interventions to promote adequate infant and young child feeding practices, identification of danger signs of malnutrition and referral to health posts and promotion of adequate hygiene, water and sanitation practices through sensitisation activities through the figure of ADECOS.
2. Preventive pharmacological activities: including semi-annual vitamin A supplementation for children 6–24 months old; semiannual deworming with albendazole for children from 12 months to 24 months of age and pregnant women from the second trimester; and malaria prophylaxis with sulfadoxine–pyrimethamine for pregnant women from 13th week of pregnancy through the health posts.

The standard of care is implemented in all the communities participating in the study, as detailed below.

#### Standard of care+nutritional supplementation

Communities allocated to this arm will receive the standard of care intervention plus nutritional supplementation to families with at least one pregnant woman enrolled. Nutritional supplementation will consist of SQ-LNS, a peanut-based ready-to-use home-fortification product to improve diet quality, for the pregnant women and their

newborn children, and a complementary food ration for their families. The rations will be as follows: an individual ration of SQ-LNS for pregnant women Enov'Mum (1 sachet of 20 g daily) and Enov'Nutributter for children over 6 months (1 sachet of 20 g daily) produced and packaged in France by Nutriset. A complementary family food ration will comprise of a basket of locally produced staple foods that complements the usual diet (300 Kcal/person/day). The caloric distribution of the basket will be 45% of cereals (corn meal—carbohydrate), 30% of legumes (beans—vegetable protein) and 25% of oil (soybean oil—fat). In addition, 1 kg of iodised salt will be provided. The food basket is distributed in order to improve adherence to the SQ-LNS, contribute to greater participation and prevent intrafamily distribution of individual rations.

### Standard of care+cash transfer

Communities allocated to this arm will receive the standard of care intervention plus an unconditional cash transfer to families with at least one pregnant woman enrolled. A total of 13 855 kwanzas/month (US$31.5, October 2022) will be delivered to households with four or more inhabitants. A total of 10 855 kwanzas/month (US$24.7, October 2022) will be delivered to households with three or fewer inhabitants. The minimum interprofessional salary in Angola at the time being is set at 35 000 kwanzas/month (US$79.5, October 2022). The amount will be directly delivered to study participants. The amounts of cash were selected based on the amount disbursed by the Kwenda Project, an Angolan Government programme funded by the World Bank aiming to fight poverty and to promote sustainable development in the communities with very poor households.

### Measurement of health outcomes/effectiveness

The outcomes of the interventions will be assessed through repeated cross-sectional household surveys at baseline, at 6 months of pregnancy, 9 months of pregnancy, 3, 6, 12, 18 months of the newborn child, and endline at 24 months of newborn child.

The MuCCUA trial will test the effect of each of the experimental interventions (standard of care+nutritional supplementation, standard of care+cash transfer arms) relative to the control (standard of care arm). A detailed description of the trial outcomes is included in the trial protocol.[14]

### Primary outcome

The MuCCUA trial primary outcome is prevalence of stunting as measured by height-for-age z-score (HAZ) or length-for-height z-score (LAZ) <−2 SD and mean HAZ or LAZ in children at 24 months of age in the provinces of Huila and Cunene.

The trial sample size was powered to measure an expected 13% reduction in stunting overall and to detect a difference in the non-inferiority margin of 10% between the proportions in the intervention groups (standard of care+nutritional supplementation and standard of care+cash transfer) and the comparison group (standard of care alone) post-intervention. We will conduct a cost-effectiveness analysis for cases of stunting averted if there is evidence of impact on the prevalence of stunting in children at 24 months at the 0.1 significance level or below. Cases averted by the two interventions relative to the control group for the outcome will be calculated at 24 months.

### Secondary outcomes

The MuCCUA trial has a number of secondary outcomes including: infant mortality rate at 3, 6, 12, 18 and 24 months, proportion of neonatal low birth weight and low birth weight for gestational age, proportion of children with anaemia, cumulative incidence of morbidity (malaria, diarrhoea and pneumonia), primary caregiver's knowledge, attitudes and practices related to perinatal and children's caring practices including breast feeding and hygiene and sanitation, and women and children minimum dietary diversity.

### Heterogeneity analyses

In addition to calculating the efficiency and cost-effectiveness of the MuCCUA trial, heterogeneity analysis will be conducted to assess how costs and impacts of the interventions are distributed among the target population. The economic evaluation will be stratified through subgroup analyses of the primary outcome based on differences of the study population characteristics. Disaggregation of the findings will be performed in subgroups per treatment arm, based on the two provinces and on the four communes of the study. The environment and way of living in each of the two provinces are slightly different in terms of, for example, distances between households' compounds and health services. In the province of Cunene, households are very far apart from each other and they rely solely on themselves not having much interaction with other members of the communes. On the contrary, in the province of Huila, communities live closer together in villages and there are more interactions that exist between them. Socioeconomic status will not be included as a subgroup as we do not expect important differences because one of the inclusion criteria to participate in the study was to experience extreme poverty. In each of these subgroups, community group discussions will be carried out with different samples of participants according to their access to basic services and distance to the distribution points of the interventions.

### Identification, measurement and valuation of resource use

The cost-efficiency and cost-effectiveness of the MuCCUA interventions will be measured from a societal perspective,[31 32] taking into account costs incurred by the programme providers, namely the Institute of Local Development-Social Support Fund of Angola (FAS) and Vall d'Hebron Institute of Research (VHIR) (programme provider costs), ADECOS and by the beneficiaries who are intervention participants and their households. Resource

**Table 1** Overview of resource use and cost measures included in the economic evaluation of the MuCCUA interventions

| Perspective/cost category | Type of costs | Description | Source | Sample size |
|---|---|---|---|---|
| Programme/implementing agency | Direct | Costs of implementing the intervention | 1. Project accounts of the implementing agencies<br>2. Interviews with the project staff | N/A |
| | Indirect | Opportunity cost of donated items, volunteer time | 1. Interviews with the project staff<br>2. Project records on volunteer involvement<br>3. Project accounts | N/A |
| Provider/health system (health providers) | Direct | Changes in demand/utilisation of nutrition and health services/costs of referrals | 1. Baseline and endline cross-sectional surveys (for information on changes in cost of health-seeking behaviour)<br>2. Project records for number of referrals | All participants from the study<br>All referrals made in intervention and control clusters |
| | Indirect | Opportunity cost of increase in workload health workers in the study area<br>Opportunity cost of the time spent by health workers participating in the intervention's meetings | Time use survey with health workers<br><br>1. Project records on health workers' participation in intervention<br>2. Project records on number of health strengthening meetings and attendants<br>3. Published reports on local wage information based on skill category | A purposive sample of health workers will be selected for time use interviews<br>N/A |
| Participants/households | Indirect | Opportunity cost of participation in the interventions, group meetings and home visits | 1. Household survey<br>2. Focus group discussions<br>3. Qualitative interviews with a subsample of participants | All participants from the study<br>A purposive sample of 6–9 participants per group discussion in each of the subgroups by intervention arm<br>A purposive sample of 1–2 participants per each of the subgroups and intervention arms |

MuCCUA, Mother and Child Chronic Undernutrition in Angola; N/A, not available.

use will be measured using financial expenditure data combined with micro-costing to allow for estimation of both financial and economic costs. All cost data will be collected prospectively from the launch of the MuCCUA community trial project for a 3-year (2022–2025) time period, and divided into start-up and recurrent costs. Table 1 provides an overview of the resource use and cost measures that are described in more detail further below.

**Programme-related costs**
Costs will be estimated using a mixed-methods approach combining costs estimated both from accounting records and using an ingredients approach.[43] Resource use will be measured using financial expenditure data combined with micro-costing to allow for estimation of both financial and economic costs of each of the three intervention arms. An activity-based costing method will be applied to allocate and estimate financial costs per major programme activity, using accounting records and staff time allocation sheets based on the SEEMS-Nutrition standardised tools.[37] The financial cost estimates will track payments, focusing on the resources coming from organisation, coordination and implementing partners

including VHIR and FAS, and expenditure data for MuCCUA trial coming from Crescer Project expense reports. In addition, country salary scales will be used to estimate costs incurred by implementation partners (municipal supervisors, community activity assistants and medical staff). The ingredients approach will be used to estimate economic costs, costs not included in accounting records and indirect costs incurred by beneficiaries; these will be collected via key informant interviews, focus group discussions, structured questionnaires, household surveys and/or observation.[44] Qualitative information will be collected through interviews from purposefully selected key informants from implementing organisations (VHIR, FAS), government officials (municipal supervisors, community activity assistants, FAS coordinators and health workers), beneficiaries and other members of the community. In order to use the activity-based costing methodology, staff time allocation will be assessed to estimate and allocate costs to activities. In addition, focus group discussions with beneficiaries will be performed to better understand opportunity costs to them and their communities from participating in the interventions. A total of 24 focus group discussions including 8–10 beneficiaries will be performed, two per trial arm in each of the four communes.

All costs will be collected prospectively from the project expense reports of the implementing and technical partners and entered into a programme costing tool in Microsoft Excel on an annual or monthly basis as appropriate. Data collection will be for the 3-year (2022–2025) time period from the launch of MuCCUA community trial. This method is recommended to assess the total costs of complex multisectoral nutrition and public health interventions that have multiple health and non-health effects.[29]

### Financial or expenditure data

Financial costs will be collected by extraction of accounting sheets provided by financial officers of implementing partners (VHIR and FAS). Data from administrative financial records will be entered in Excel worksheets and assigned cost category codes for activities, inputs, thematic areas and timing of activities by implementation partners. For capital and start-up costs occurring at the start of the project, we will adjust financial records to reflect both financial and economic costs. Data obtained from interviews will be used to estimate averages and ranges of project-related time use by each type of implementing partner, and for project beneficiaries. We will estimate the total time for implementing partner and beneficiaries and value it with either the appropriate civil servant cost per minute (for government workers) or use a prevailing wage rate for rural or urban staff and beneficiaries. We will add in average costs of out-of-pocket expenses that have not been covered by the project. Labour and supply costs will then be coded (allocated) to the specific project activities and combined with the financial expenditure data to obtain total programme financial and economic

costs. Capital and start-up costs will be annualised over their expected useful life.[45]

### Donated items and opportunity costs

Some resources, such as donated items and volunteer time, are not routinely captured in project accounting records and will need to be identified and converted to economic costs using their market value.[45–47] Common donated items are equipment or other capital items donated or owned by implementing agency. These will be identified through key informant interviews with the project staff.

Most of the volunteer time is related to designing messages for the standard of care intervention, where several meetings were held with experts who were volunteers. Detailed information on the number of meetings, their duration and participants is being documented by the project. The opportunity cost of the time invested by the experts will be estimated as a proportion of their salary or a salary equivalent using published national/local wage rate reports. In addition, some community members volunteer for the project mainly in sensitisation activities. The opportunity cost of volunteer time donated to programme implementation will be estimated using published national/local wage rate reports.

### Public health provider cost/provider or health system cost

MuCCUA interventions are likely to increase demand for care services provided by community frontline health and nutrition workers such as ADECOS, community activity assistants and midwives, and other healthcare providers, such as malnutrition treatment centres or nutrition rehabilitation centres and primary health structures in the study area. In addition, there is a time cost of direct involvement in the project activities for the frontline workers and healthcare providers.

### Cost of changes in utilisation of health and nutrition providers

Changes in the utilisation of health and nutrition services may occur in the form of increased demand as risk of malnutrition and other child illnesses are better identified in the communities by ADECOS. These changes in service provided by healthcare providers will be collected through questionnaires administered from study participants in both intervention and control areas at the MuCCUA endline household survey. The questionnaires ask participants on the frequency of attendance to the health posts, at baseline and at endline, and results will be compared before and after, and between arms.

### Opportunity costs of MuCCUA activities/health workers

The MuCCUA interventions may increase the workload of health workers by increasing demand for their services. In contrast, the new community-based workers (or ADECOS) introduced by the MuCCUA trial, coupled with a stronger emphasis on preventive activities, may decrease the workload of existing health workers in the study areas. In addition, project activities include sensitisation and training sessions for healthcare providers in

maternal, infant and young child health preventive activities in the intervention and control arms. Information on the number of meetings, their duration and participation is being documented by the project. The opportunity cost of involvement in the standard of care intervention or the value of the time spent in training by health workers will be measured as a proportion of their salary, using publicly available data on their salaries.

### Participants and their household costs

MuCCUA interventions may influence participants and their households' costs in a number of ways. Impacts on beneficiaries and their families may include changes in household food and non-food consumption patterns and spending (regarding variety, quantity or quality of food), changes in health-seeking behaviours and associated costs as well as time cost of participation in the interventions and of attending group meetings and home visits.

#### *Opportunity cost of participating in the interventions*

Participating in the MuCCUA interventions will incur some direct and indirect costs to beneficiaries and their families. These costs may include the cost of travelling to the health posts, food and cash distributions sites as well as the opportunity cost of time spent travelling. Basic travel time and costs will be measured using survey questions to all study participants during household surveys. In addition, the cost of attending community group meetings and engaging in household visits performed by the ADECOS will be estimated by collecting time allocation data from the individuals attending the group meetings at several household surveys. Focus group discussions with a subsample of participants purposively selected and equally distributed across the intervention and control arms will also be undertaken to explore any other opportunity costs to households and their communities of the implementation of the interventions. All these costs will feed into the overall cost analysis.

### Cost-efficiency analysis

Total costs will be estimated for each of the three arms of the MuCCUA trial: standard of care, standard of care+nutritional supplementation and standard of care+cash transfers. We will estimate the cost-efficiency of MuCCUA trial by combining activity-specific cost data with routine data on programme outputs, in order to calculate the cost per output achieved. Numbers of index pregnant women, their newborn children until they are 24 months old and households reached by the intervention will be tallied by programme staff. Ratios of cost per woman, child and household reached will be calculated by dividing the total estimated intervention costs by the respective numbers reached.

### Cost-effectiveness analysis

Economic evaluation will be conducted as a within-trial analysis using the intention-to-treat results, and will be presented in terms of incremental cost-effectiveness ratios (ICERs), calculated as the difference in total costs of standard care only (or comparator) versus nutritional supplementation or cash transfer divided by the difference in mean effects of each intervention versus control. As mentioned previously, ICERs will be calculated in terms of cost per stunting case prevented among children at 24 months old if there is a statistically significant difference in the outcome between intervention study arms.

The ICER will be calculated to compare the additional costs per improved outcome achieved in interventions, including nutritional supplements or cash transfers, compared with the standard of care intervention. Where feasible, we will disaggregate costs for different time frames, including start-up versus recurrent costs.

Sensitivity analysis will be performed to determine the extent to which the results of the analysis might change given plausible variation in study parameters related to cost drivers and stunting rates. To obtain an estimate of how costs could vary, a range of costs will be calculated based on different scenarios such as local costs scenario, product price or coverage. Univariate and multivariate probabilistic sensitivity analyses will assess effect of the main cost drivers on the results.

All costs will be presented in 2022 Angola kwanzas and in US dollars (USD). We will use the average monthly exchange rate for the year of cost data collection to convert costs into USD. Any costs encountered in the past will be inflated using the local gross domestic product deflator of Angola, before converting to USD. In addition, some expenses will be in euros and we will use the average exchange rate for the year of cost data collection to convert costs into USD. All costs will be adjusted for inflation using the Angolan Consumer Price Index and will be converted to 2022 USD using the purchasing power parity conversion factor for Angola. Moreover, the costs will be discounted using a standard discount rate of 3%, as recommended by WHO-CHOICE[48] and the Gates/IDSi Reference Case for Economic Evaluation.[49]

## CONCLUSION

The findings of this trial will provide evidence on which intervention package can have the greatest impact on preventing stunting in children under 2 years in Angola and improve comparability with other contexts. Robust understanding of the costs and benefits of these types of nutrition strategies is critical for priority setting and to motivate ongoing donor investment and government partnership.

The cost estimation of the different arms of the trial will complement the effectiveness results to provide useful information to the Government of Angola and other policymakers on the most cost-effective and cost-efficient intervention to prevent stunting. Additionally, this study aims to gain understanding on the cost drivers, to promote inclusion of economic evaluations

in other nutrition intervention studies and to improve comparability with other contexts.

One of the limitations of the MuCCUA trial is that it is only powered to test the differences between each of the intervention arms and the comparison group, but not to test the differences between the two intervention arms. The latter would require a larger sample size and resources beyond those available for the trial. However, the possibility of a direct comparison between the interventions will be explored. The protocol has all necessary ethical approvals and the findings of this study will be disseminated within academia and the wider policy sphere.

**Author affiliations**
¹Centro Nacional de Medicina Tropical, Instituto de Salud Carlos III, Madrid, Spain
²Escuela Internacional de Doctorado, Universidad Nacional de Educación a Distancia, Madrid, España
³CIBER Enfermedades Infecciosas, ISCIII, Madrid, Spain
⁴Infectious Diseases Department, Vall d'Hebron University Hospital, Barcelona, Spain
⁵Centro Nacional de Epidemiologia, Instituto de Salud Carlos III, Madrid, Spain
⁶CIBER Epidemiología y Salud Publica (CIBERESP), Madrid, Spain
⁷Fundo Apoio Social-Local Development Institute, Luanda, Angola
⁸Faculdade de Medicina da Universidade Mandume Ya Ndemufayo, Lubango, Huíla, Angola
⁹Action Against Hunger Spain, Madrid, Spain
¹⁰Stony Brook University Program in Public Health, Stony Brook, New York, USA

**Acknowledgements** We would like to acknowledge the MuCCUA Study participants and field teams including ADECOS, their supervisors and students of the Universidade Mandume ya Ndemufayo for their collaboration in the data collection process. Furthermore, to the Crescer consortium; the FRESAN Programme implementers; municipal and province administrations of communes and Huila and Cunene provinces; and the health, food and nutrition security and social action offices of the National Government of Angola.

**Contributors** RM-C and EC designed the protocol for the economic evaluation and prepared the first draft of the paper. ET, MR-B, ZH, IA, FR, LMF, AS-G, EI, AV, TM, AG, JCL, IM and EC contributed to the conception and design of the MuCCUA trial protocol in which this study is embedded. RM-C, EC and CP designed the cost data collection tools and the processes for allocating costs. ET, IA, FR, LMF and AS-G led the supervision of all intervention implementation activities and contributed to field data acquisition. All authors contributed to the review of this manuscript and provided comments. All authors have read and approved the submitted manuscript.

**Funding** The Crescer Project is funded by the European Union EuropeAid contract no FED/2020/418-106 and co-financed by the Crescer consortium partners. The MuCCUA trial is funded by the European Union as a part of the Crescer Project (IV component of the FRESAN Programme) for the 4-year duration of the project.

**Competing interests** None declared.

**Patient and public involvement** Patients and/or the public were involved in the design, or conduct, or reporting, or dissemination plans of this research. Refer to the Methods section for further details.

**Patient consent for publication** Not required.

**Provenance and peer review** Not commissioned; externally peer reviewed.

**ORCID iDs**
Rocio Martin-Cañavate http://orcid.org/0000-0002-3719-668X
Estefania Custodio http://orcid.org/0000-0002-5514-3151
María Romay-Barja http://orcid.org/0000-0002-0177-6885

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
