## [Reviewer comments · BMJ Open]

ARTICLE DETAILS

TITLE (PROVISIONAL)	Protocol for the economic evaluation of a three-arm community cluster randomized controlled trial to prevent chronic malnutrition in children under two years in rural Angola (MuCCUA trial)
AUTHORS	Martin-Cañavate, Rocio; Custodio Cerezales, Estefania; Trigo, Elena; Romay-Barja, María; Herrador, Zaida; Aguado, Isabel; Ramirez, Ferran; Faria, Lourdes Maria; Silva-Gerardo, Ana; Lima, Jose Carlos; Iráizoz, Eva; Marques, Tayná; Vargas, Antonio; Gomez, Amador; Puett, Chloe; Molina, Israel

VERSION 1 – REVIEW

REVIEWER	Maffioli, Elisa University of Michigan
REVIEW RETURNED	17-May-2023

GENERAL COMMENTS	Protocol for the economic evaluation of a three-arm community cluster randomized controlled trial to prevent chronic malnutrition in children under two years in rural Angola (MuCCUA trial) The protocol is well structured and it describes the trial and analysis concisely and in a clear way. The authors also attached a detailed protocol of the trial which is also helpful to get additional information. The trial registration on clinicaltrials.gov is also key to compare. I hope these comments are helpful to improve the protocol. 1) Description of the trial. The three interventions are described well in the trial design, but could be improved in other parts of the text. I am a bit confused about defining standard of care as an intervention. While this is not a "pure" control, this represents the status quo. Is the research team implementing any additional "intervention" or assuring that the standard of care is of a certain quality? I know the terminology needs to be aligned with what was registered, but it quite misleading. If the authors could review the text to clarify what is comparator vs. what are the two interventions that would be helpful for a reader who will only look at this protocol. For example, in the abstract it is not clear whether the 3 intervention arms are separate interventions or some combination of those. The term "combining" is quite misleading as it suggests all of 3 interventions are combined, but in practice one arm is the status quo, the two others are interventions that are added to the status quo. There is no combination of cash and nutritional supplements.
---

	For example, in Page 5 Row 15 "two interventions" are mentioned. It would be helpful to define which ones, as only at this point in the text it is clear that there are two treatment arms and one control group. In the MUCCUA interventions description, it would also be helpful to explain how the basket of food and cash amount were chosen, and why (e.g., did they satisfy certain nutrition requirements? Based on which evidence?) 2) Contribution and importance of economic analysis The authors describe very briefly the evidence around similar interventions, but there is quite a lot of existing evidence, especially around cash transfers. It might be helpful for the reader to extend the paragraph that describes how these two interventions (nutrition supplementation with SQ-LNS and family food ration and unconditional cash transfers) are effective for nutrition, separately or together. Maybe this is done in [11] but a longer paragraph than rows 43-54 could be helpful for the reader to understand the importance of the trial and of the consequent economic analysis. Some references Levere, Michael, Gayatri Acharya, and Prashant Bharadwaj, "The Role of Information and Cash Transfers on Early Childhood Development, Evidence from Nepal," 2016 Carneiro, Pedro, Lucy Kraftman, Giacomo Mason, Lucie Moore, Imran Rasul, and Molly Scott, "The impacts of a multifaceted prenatal intervention on human capital accumulation in early life," American Economic Review, 2021, 111 (8), 2506–49. The authors also describe that more evidence is needed on "costs, cost-effectiveness, and benefits". First, this sentence is a bit confusing since cost-effectiveness analysis incorporates costs and benefits. I would suggest to rephrase the sentence. Could the authors better explain what are the gaps in the literature? Why do we need a cost-efficiency and cost-effectiveness analysis of this type of trial? Which evidence exist on cost-effectiveness of nutrition-related interventions? Why this type of analysis is not conducted in other trials? What is new about this study? For example, some studies do not conduct CEA because they lack the data. Other studies do a different type of analysis (Internal Rate of Return), see Carneiro et al. 2021. However some evidence exists. A lot of the discussion in the protocol is about how to compare across interventions. However, 1) there are a lot of RCT that conduct multifaceted interventions; 2) Cost-effectiveness analysis could be conducted if data are collected and outcomes can be measured in the same way (e.g. stunting which is usually measured in nutrition-related interventions). It would be helpful to connect the existing literature (point above) and better explain how this study will contribute and improve on the existing evidence. 3) Heterogeneity analysis The authors describe the "equity impact analysis". However, this seems to be simply an heterogeneous analysis. The "equity term" is also a bit misleading. Or are the authors interested in examining equity effects in CEA? See: Avanceña ALV, Prosser LA. Examining Equity Effects of Health Interventions in Cost-
--	--

	Effectiveness Analysis: A Systematic Review. Value Health. 2021 Jan;24(1):136-143. doi: 10.1016/j.jval.2020.10.010. Epub 2020 Dec 3. PMID: 33431148. There is no enough clarity on the equity/heterogeneity analysis. Which characteristics of the population will be considered and why? Pg7. The equity impact analysis is mentioned for the first time and should be better explained. Analysis It would be helpful to better align secondary outcomes registered for the trial with the secondary outcomes used for the analysis. It seems the authors only consider prevalence of moderate stunting (<-2SD) and not severe stunting. Why? I also doubt the trial is powered for mortality effects, so I am concerned about calculating cost per mortality case averted among children at 24 months old. Could the authors clarify if they have power for secondary outcomes? Is the sample size powered to detect a 13% or 10% reduction in stunting? Please align explanation with 11.1. Calculation of size sample in the attached protocol. Is the "prevalence of chronic malnutrition" defined as moderate or severe stunting? Other comments I wonder how "pregnant women and families were intimately involved in the implementation of the intervention". This sounds like they had a say in how the intervention was rolled-out which seems surprising since this is a research study. Could the authors explain this sentence or rephrase? It would be helpful to add which type of sensitivity analysis will be conducted, if any. Be consistent with numbers (either 1 decimal point or round): e.g. stunting 49.9%/50% 37.2%/37% in the abstract and introduction.
--	---

REVIEWER	Adams, Katherine University of California Davis, Institute for Global Nutrition
REVIEW RETURNED	01-Sep-2023

GENERAL COMMENTS	Review of Protocol for the economic evaluation of a three-arm community cluster randomized controlled trial to prevent chronic malnutrition in children under two years in rural Angola (MuCCUA trial) This paper describes the protocol for conducting an economic analysis to assess the cost-efficiency and cost-effectiveness of different strategies for addressing malnutrition compared to the standard of care in Angola. The economic analysis will be nested within a cluster RCT. In general, the economic analysis seems well planned. I do have a number comments, but most of them are fairly minor requests for clarification. Note that there were two different sets of page numbers on each page of the draft paper. I used the numbers at the bottom of each page in my comments below.
--

Page 2 line 35: The sensitivity analysis is never described in the main protocol. Please add a paragraph or two describing how the sensitivity analysis will be conducted (e.g., by varying which variables, and informed by what?).

Page 3 strengths and limitations: You might consider adding as a limitation that the economic analyses were conducted in the context of a research project, and sometimes costs can be influenced by the research component in ways that cannot be seen/corrected for. This is true of all economic analyses embedded in research projects, but still important to point this out, I think.

Page 4 lines 17-28: Since the referenced time period has passed by quite a while, might consider changing the tense and shifting the language from “will likely worsen during the lean season” since, presumably, the specific lean season in question has also passed.

Page 5 line 10: Implements or will implement? Given that this is a study protocol and the data collection will be prospective, I assume the latter?

Page 6 lines 32-27: Please be more explicit about why conducting the costing study from a societal perspective will address all of the comparison problems identified above.

Page 7 line 9: Is cost per community event conducted really an “output” of these interventions? In this case, I think that’s more of an input (and will likely be directly costed as an activity within each intervention, right)?

Page 7 line 47: Typo – “they were they”

Page 7 lines 48-49: The participants were intimately involved in the implementation of the intervention? In what ways, specifically? This seems very odd. Also, should be interventions rather than intervention for consistency with the plural use above.

Page 7, lines 57-60: Please describe exactly how the participants and families will disseminate baseline information (and what baseline information will they be disseminating?). It seems like this aspect of the trial might harm the external validity of the impacts/outcome estimates. Would this type of activity by participants and their families be expected to also happen in a real-world program?

Page 8 lines 30-32: Why don’t you expect it?

	Page 8 line 30: You don't really "belong" to very low levels of poverty. You might reword as something like "experience extreme poverty" or something like that. Page 8 line 56: Your definition already includes sharing housing and food. What would be examples of other vital shared needs? Page 10 line 33-40: Confirming that IFA supplementation for pregnant women is not part of the standard of care in Angola? Page 11 line 19: Which is approx. how many USD per month? And how does this compare to the minimum wage (or another similar metric) in Angola? Page 12 lines 12-23: Will cost-effectiveness be evaluated on the basis of each of these secondary outcomes as well as stunting at 24mo? Would be helpful to be explicit about this. Page 12 line 35: I'm not sure any population characteristics were "mentioned earlier". Not clear what characteristics you are referring to here. Page 12 lines 51-58: Earlier you noted you would measure the costs faced by "programme, provider and households". Here you describe implementing agencies and users of the interventions. Would be helpful to adopt one set of terms and use them consistently with respect to costs to whom. Page 15 lines 35-40: Please describe a bit more about what this qualitative data collection will involve and how it will be used to inform cost estimates. Page 16 lines 46-47: If specific messaging was developed for the "Standard of Care" components of the interventions, it's not really SOC, right? Anything that is new/different from what normally takes place would be in addition to the SOC. Please clarify. Page 17 lines 28-31: It's unclear how information collected from study participants could be used to measure changes in service utilization. Do you mean differences in service utilization between the SOC arms and the other intervention arms? It seems like before/during service utilization records for relevant health facilities would also be important, if such records exist? Page 17 lines 37-38: How will increased workloads be "costed"? Will they work extra time at their wage rate? Or will existing staff be expected to shoulder an increased workload without compensation? If the latter, even if this is not directly accounted for in the costing, might be important to consider impacts of motivation/mental health/performance.
--	--

	Pag 17 lines 44-48: Again, it's unclear how new training and sensitization can be considered SOC... Page 19 lines 11-12: But only if there is a statistically significant difference in these outcomes between SOC and the other intervention arms, right? Page 19 lines 10-20: As I mentioned above, I think important to note the limitation that that the economic analysis was conducted in the context of a research project, and identifying and netting out research vs implementation costs can sometimes be difficult.
--	--

VERSION 1 – AUTHOR RESPONSE

REVIEWER 1

Protocol for the economic evaluation of a three-arm community cluster randomized controlled trial to prevent chronic malnutrition in children under two years in rural Angola (MuCCUA trial)

The protocol is well structured and it describes the trial and analysis concisely and in a clear way. The authors also attached a detailed protocol of the trial which is also helpful to get additional information. The trial registration on *clinicaltrials.gov* is also key to compare. I hope these comments are helpful to improve the protocol.

Dear reviewer, we appreciate your thorough revision of our manuscript and the questions you raised. They have triggered internal discussions and careful revision of concepts and procedures. We have addressed them all one by one and we have numbered all of them for easiness of communication. We believe the document has improved substantially and we hope content has gained in clarity for you as well. Please be aware that the text page numbers indicated in the responses correspond to the Main manuscript marked copy document.

1) **Description of the trial.** The three interventions are described well in the trial design, but could be improved in other parts of the text.

1. I am a bit confused about defining standard of care as an intervention. While this is not a "pure" control, this represents the status quo. Is the research team implementing any additional "intervention" or assuring that the standard of care is of a certain quality? I know the terminology needs to be aligned with what was registered, but it quite misleading. If the authors could review the text to clarify what is comparator vs. what are the two interventions that would be helpful for a reader who will only look at this protocol.

We apologise for the confusion and we provide more background information to help clarify the design and different arms of the trial.

The standard of care is not exactly the status quo in Angola. In 2014, the Ministry of Territorial Development of Angola designed a new community development policy aiming to address human resource shortages in remote areas effectively, in order to provide primary and preventive care, and to improve health status and quality of life in rural communities. This policy is to be implemented by the Angolan Institute of Local Development- Social Support Fund (FAS) and includes the recruitment of two new figures: the sector supervisors of the main extension services (health, social protection, and agriculture), and the ADECOS (Community and Health Development Agents). The ADECOS are

trained in the identification of danger signs of malnutrition and referral to a health post, appropriate WASH and dietary practices and work with the families and communities as linkages with the health system (Ministério da Saúde, República de Angola. Política Nacional de Agentes de Desenvolvimento Comunitário e Sanitário (ADECOS), Luanda, Angola 2014)..

In 2020, at the time the MuCCUA trial was designed, the policy was at different stages of implementation across the country, but the Southern provinces had not yet started its implementation at the health care level. Therefore, in the study areas of the provinces of Huila and Cunene, the Crescer project supported the policy by funding the recruitment of ADECOS through the FAS (which is a consortium partner of the Crescer project).

The standard of care in the MuCCUA trial comprises the recruitment of ADECOS and the community activities they develop, as well as the delivery of preventive pharmacological products through the health posts. The project team selected these activities from those that were described in the National Development Plan of Angola and the Angola Multisectoral Strategic Plan on Nutrition, and are in line with the WHO maternal and infant health care recommendations. Therefore, the research team in this trial is assuring the implementation of the ADECOS, securing their subsidies and the quality of some other aspects of the standard of care such as providing or securing the stock of deworming, vitamin A and malaria prophylaxis in the health posts. Trainings and sensitization are done to ensure quality and they are a part of the current ADECOS policy.

The standard of care is provided to all communities participating in the MuCCUA trial. The arm that only receives the standard of care is considered the comparator, as the other two arms include the standard of care plus another intervention, either the unconditional cash transfer or the nutritional supplementation.

We have amended the text in order to make it more clear (page 5 and page 12).

2. For example, in the abstract it is not clear whether the 3 intervention arms are separate interventions or some combination of those. The term "combining" is quite misleading as it suggests all of 3 interventions are combined, but in practice one arm is the status quo, the two others are interventions that are added to the status quo. There is no combination of cash and nutritional supplements.

That is correct there is no combination of cash and nutritional supplements within the same arm. However, all interventions include the standard of care (SOC). The three arms are: SOC alone, SOC + cash transfers, and SOC + nutritional supplementation. We have modified the text accordingly.

3. For example, in Page 5 Row 15 "two interventions" are mentioned. It would be helpful to define which ones, as only at this point in the text it is clear that there are two treatment arms and one control group.

Agree. We have modified the text accordingly.

4. In the MUCCUA interventions description, it would also be helpful to explain how the basket of food and cash amount were chosen, and why (e.g., did they satisfy certain nutrition requirements? Based on which evidence?)

The objective of the food basket distribution is to improve the adherence to the SQ-LNS, contribute to greater participation and prevent intra-family distribution of individual rations. Thus, it provides complementary calories to the family diet (around 300 Kcal per person and per day) but does not replace it. The composition of the food basket was calculated based on the macronutrients requirements of carbohydrates, protein and lipids and the recommended percentage distribution of the nutrients in a healthy diet.

The cash amount was decided based on a cash transfer project implemented by the World Bank in these provinces (the Kwenda project). The amount given is between 11 000 and 14 000 kwanzas per month and the minimum monthly salary in Angola are 35 000 kwanzas. We have added this information in the text in page 14 to help contextualize the figures.

2) Contribution and importance of economic analysis

5. The authors describe very briefly the evidence around similar interventions, but there is quite a lot of existing evidence, especially around cash transfers. It might be helpful for the reader to extend the paragraph that describes how these two interventions (nutrition supplementation with SQLNS and family food ration and unconditional cash transfers) are effective for nutrition, separately or together. Maybe this is done in [11] but a longer paragraph than rows 43-54 could be helpful for the reader to understand the importance of the trial and of the consequent economic analysis. Some references.

Levere, Michael, Gayatri Acharya, and Prashant Bharadwaj, "The Role of Information and Cash Transfers on Early Childhood Development, Evidence from Nepal," 2016

Carneiro, Pedro, Lucy Kraftman, Giacomo Mason, Lucie Moore, Imran Rasul, and Molly Scott, "The impacts of a multifaceted prenatal intervention on human capital accumulation in early life," American Economic Review, 2021, 111 (8), 2506–49.

Thank you for this comment. We have extended these paragraphs including references suggested as well as recent systematic reviews on the subject. See pages 6 and 7.

6. The authors also describe that more evidence is needed on "costs, cost-effectiveness, and benefits". First, this sentence is a bit confusing since cost-effectiveness analysis incorporates costs and benefits. I would suggest to rephrase the sentence. Could the authors better explain what are the gaps in the literature? Why do we need a cost-efficiency and cost-effectiveness analysis of this type of trial? Which evidence exist on cost-effectiveness of nutrition-related interventions? Why this type of analysis is not conducted in other trials? What is new about this study? For example, some studies do not conduct CEA because they lack the data. Other studies do a different type of analysis (Internal Rate of Return), see Carneiro et al. 2021. However some evidence exists.

Yes, we agree. We have extended this paragraph also including evidence from recent reviews. See pages 6 and 7.

7. A lot of the discussion in the protocol is about how to compare across interventions. However, 1) there are a lot of RCT that conduct multifaceted interventions; 2) Cost-effectiveness analysis could be conducted if data are collected and outcomes can be measured in the same way (e.g. stunting which is usually measured in nutrition-related interventions). It would be helpful to connect the existing literature (point above) and better explain how this study will contribute and improve on the existing evidence. Agree. The heterogeneity in terms of effective and costs measurement is one of the issues at stake regarding gaps on nutrition multisectoral programs. The tools that the SEEM-Nutrition initiative has developed try to minimize it. We have followed the SEEM guide in order to provide results comparable and useful to the rest of the community. New have added it to the manuscript. Page 8.

3) Heterogeneity analysis

8. The authors describe the "equity impact analysis". However, this seems to be simply an heterogenous analysis. The "equity term" is also a bit misleading. Or are the authors interested in examining equity effects in CEA? See: *Avanceña ALV, Prosser LA. Examining Equity Effects of Health Interventions in Cost-Effectiveness Analysis: A Systematic Review. Value Health. 2021 Jan;24(1):136-143. doi: 10.1016/j.jval.2020.10.010. Epub 2020 Dec 3. PMID: 33431148.*
9. Yes, we were actually referring to the heterogeneity/ subgroup analysis and we have modified it and explained it in further detail in the text, pages 9 and 15 "heterogeneity analysis will be

conducted to assess how costs and impacts of the interventions are distributed among different subgroups of the target population.” The subgroups are based in geographical location (province and commune level) and access to services of the communities.

10. There is not enough clarity on the equity/heterogeneity analysis. Which characteristics of the population will be considered and why?

As commented above we will perform heterogeneity analysis based on province, commune and access to services that are mentioned in page 15. The environment and ways of living in each of the two provinces are slightly different in terms of distances between household's compounds and to health services. For example, in the province of Cunene, households called “kimbos” are very far apart from each other and they rely solely on themselves while in the province of Huila, communities live closer together in villages and there are more interactions between them.

11. Pg7. The equity impact analysis is mentioned for the first time and should be better explained. Thank you for pointing that out. The explanation is given in the two comments above.

Analysis.

12. It would be helpful to better align secondary outcomes registered for the trial with the secondary outcomes used for the analysis. It seems the authors only consider prevalence of moderate stunting (<-2SD) and not severe stunting. Why?

The only outcome that would be included in economic analysis is global stunting defined as height-for-age z-score <-2 SD which includes both moderate and severe stunting. The text has been modified accordingly.

13. I also doubt the trial is powered for mortality effects, so I am concerned about calculating cost per mortality case averted among children at 24 months old. Could the authors clarify if they have power for secondary outcomes?

That is correct; the study is only powered for stunting effects. However, secondary outcomes will be assessed for interpretation purposes of intermediate outcomes that could explain impacts on stunting.

14. Is the sample size powered to detect a 13% or 10% reduction in stunting? Please align explanation with 11.1. Calculation of size sample in the attached protocol. Is the “prevalence of chronic malnutrition” defined as moderate or severe stunting?

The sample size is powered to detect a 13% reduction in stunting overall, but to detect a difference in the non-inferiority margin between proportion in 10% with 80% power. We have added this to the text. And the prevalence of chronic malnutrition is defined as overall stunting defined as Height for Age Z score below -2 that includes both moderate and severe forms of stunting.

Other comments

15. I wonder how “pregnant women and families were intimately involved in the implementation of the intervention”. This sounds like they had a say in how the intervention was rolled-out which seems surprising since this is a research study. Could the authors explain this sentence or rephrase?

We agree that the sentence was misleading and have rephrased it as follows, in pages 9 and 10. “Pregnant women and their families were not involved in setting the research question or the outcome measures. The project is promoting understanding and dissemination of key messages around the interventions to motivate community involvement during and beyond the study. A feedback/suggestions mechanism for participants and communities has been implemented as well. At the end of the study, results will be shared and discussed with participants, their families and communities through dedicated workshops. Results and all the knowledge generated should promote change at community, regional and national level in policies and practices.”

16. It would be helpful to add which type of sensitivity analysis will be conducted, if any.

We included the explanation missing of the sensitivity analysis in page 22. "Sensitivity analysis will be performed to determine the extent to which the results of the analysis might change given plausible variation in study parameters related to costs drivers and stunting rates. To obtain an estimate of how costs could vary a range of costs will be calculated based on different scenarios such as local costs scenario, products price or coverage. Univariate and multivariate probabilistic sensitivity analyses will assess effect of the main cost drivers on the results."

17. Be consistent with numbers (either 1 decimal point or round): e.g. stunting 49.9%/50% 37.2%/37% in the abstract and introduction. Thanks for the comment. Agree. We have harmonized all numbers to only one decimal point.

Reviewer 2

Review of Protocol for the economic evaluation of a three-arm community cluster randomized controlled trial to prevent chronic malnutrition in children under two years in rural Angola (MuCCUA trial)

This paper describes the protocol for conducting an economic analysis to assess the cost-efficiency and cost-effectiveness of different strategies for addressing malnutrition compared to the standard of care in Angola. The economic analysis will be nested within a cluster RCT. In general, the economic analysis seems well planned. I do have a number comments, but most of them are fairly minor requests for clarification. Note that there were two different sets of page numbers on each page of the draft paper. I used the numbers at the bottom of each page in my comments below.

Dear reviewer, thank you for your feedback and for your clarification on page numbering. Your comments have triggered valuable discussions within the team. We have addressed all comments one by one and we have numbered all of them for easiness of communication. We believe the document has improved substantially and we hope content has gained in clarity for you as well. Please be aware that the text page numbers indicated in the responses correspond to the Main manuscript marked copy document.

1. Page 2 line 35: The sensitivity analysis is never described in the main protocol. Please add a paragraph or two describing how the sensitivity analysis will be conducted (e.g., by varying which variables, and informed by what?).

Thank you for pointing this out. We have added a paragraph giving more details in page 22.

"Sensitivity analysis will be performed to determine the extent to which the results of the analysis might change given plausible variation in study parameters related to costs drivers and stunting rates. To obtain an estimate of how costs could vary a range of costs will be calculated based on different scenarios such as local costs scenario, products price or coverage. Univariate and multivariate probabilistic sensitivity analyses will assess effect of the main cost drivers on the results."

2. Page 3 strengths and limitations: You might consider adding as a limitation that the economic analyses were conducted in the context of a research project, and sometimes costs can be influenced by the research component in ways that cannot be seen/corrected for. This is true of all economic analyses embedded in research projects, but still important to point this out, I think.

Thank you this comment. We have included the limitation in page 3 "The economic evaluation will be conducted in the context of a research project, and costs could be influenced by the research component in ways that cannot be seen or corrected for." In addition, the strengths and limitations section was shortened as suggested by the editor.

3. Page 4 lines 17-28: Since the referenced time period has passes by quite a while, might consider changing the tense and shifting the language from "will likely worsen during the lean season" since, presumably, the specific lean season in question has also passed.

Agree. The text has been modified. See page 4.

4. Page 5 line 10: Implements or will implement? Given that this is a study protocol and the data collection will be prospective, I assume the latter?

The trial has already started but the cost data collection will be prospective that is why we keep implements.

5. Page 6 lines 32-27: Please be more explicit about why conducting the costing study from a societal perspective will address all of the comparison problems identified above.

The phrase was wrongly worded. We have rephrased as follows. "The proposed economic evaluation of the MuCCUA trial aims to address the challenges mentioned above by assessing the cost-effectiveness of the interventions. The perspective used will be societal, including all costs of the interventions regardless of who incurs them."

6. Page 7 line 9: Is cost per community event conducted really an "output" of these interventions? In this case, I think that's more of an input (and will likely be directly costed as an activity within each intervention, right)?

We agree with the reviewer that the cost per community event conducted is more of an input than an output. We have removed the examples in the text and we have specified exact units in page 8. "To estimate the cost-efficiency of each of the three interventions as cost per woman, child, and household reached".

7. Page 7 line 47: Typo – "they were they". Ammended.

8. Page 7 lines 48-49: The participants were intimately involved in the implementation of the intervention? In what ways, specifically? This seems very odd. Also, should be interventions rather than intervention for consistency with the plural use above. We agree with the reviewer, that this sentence was misleading. We have modified the text in page 9. "Pregnant women and their families were not involved in setting the research question or the outcome measures. The project is promoting understanding and dissemination of key messages around the interventions to motivate community involvement during and beyond the study. A feedback/suggestions mechanism for participants and communities has been implemented as well. At the end of the study, results will be shared and discussed with participants, their families and communities through dedicated workshops. Results and all the knowledge generated should promote change at community, regional and national level in policies and practices."

9. Page 7, lines 57-60: Please describe exactly how the participants and families will disseminate baseline information (and what baseline information will they be disseminating?). It seems like this aspect of the trial might harm the external validity of the impacts/outcome estimates. Would this type of activity by participants and their families be expected to also happen in a real-world program?

We have reworded this paragraph in page 9 as commented above. Baseline results will not be disseminated to participants because as the reviewer pointed out it could harm external validity of the results. Only at the end of the study, results will be shared and discussed with participants, their families and communities through dedicated workshops.

10. Page 8 lines 30-32: Why don't you expect it?

The ethnic groups of the participants of the study in the province of Huila are the Nganguela, Ovimbundu y Nyaneca, and the Kwanyamas, Nyanecas and Mundimbas in the province of Cunene. They are all sedentary and, to our knowledge, don't have specific characteristics in terms of beliefs, behaviours or practices that can affect access or disposition to participate in the interventions.

11. Page 8 line 30: You don't really "belong" to very low levels of poverty. You might reword as something like "experience extreme poverty" or something like that. Agree. We have modified the text as suggested. "The socioeconomic status of clusters and households is expected to be homogeneous, as one of the clusters inclusion criteria was to be placed in the category of extreme poverty according to the government classification"

12. Page 8 line 56: Your definition already includes sharing housing and food. What would be examples of other vital shared needs?

We defined households in order to differentiate between households and "kimbos" which are a group of households living in the same fenced space but each of them with their own roof and kitchen. Sometimes large families with different households live together in one kimbo. Therefore, "A household was considered to be the person or group of persons, with or without kinship relationships, who have been habitually living under the same roof for at least 6 months or less but with the intention of staying in the residence for the next 6 months and share food and/or other vital needs." Other vital shared needs can refer to water, cooking and eating utensils and hygiene products for example. These examples have now been added to the text in page 11.

13. Page 10 line 33-40: Confirming that IFA supplementation for pregnant women is not part of the standard of care in Angola?

The Angolan National Multisectoral Strategic Plan on Nutrition includes IFA supplementation for women 15-49 years as a recommended intervention in the community, but this one is not included in the standard of care provided by the MuCCUA trial, as explained in the response to the comment #1 of Reviewer 1 copied below.

" The standard of care is not exactly the status quo in Angola. In 2014, the Ministry of Territorial Development of Angola designed a new community development policy aiming to address human resource shortages in remote areas effectively, in order to provide primary and preventive care, and to improve health status and quality of life in rural communities. This policy is to be implemented by the Angolan Institute of Local Development- Social Support Fund (FAS) and includes the recruitment of two new figures: the sector supervisors of the main extension services (health, social protection, and agriculture), and the ADECOS (Community and Health Development Agents). The ADECOS are trained in the identification of danger signs of malnutrition and referral to a health post and appropriate WASH and dietary practices and work with the families and communities as linkages with the health system (Ministério da Saúde, República de Angola. Política Nacional de Agentes de Desenvolvimento Comunitário e Sanitário (ADECOS), Luanda, Angola 2014)..

In 2020, at the time the MuCCUA trial was designed, the policy was at different stages of implementation across the country, but the Southern provinces had not yet started its implementation at the health care level. Therefore, in the study areas of the provinces of Huila and Cunene, the Crescer project supported the policy by funding the recruitment of ADECOS through the FAS (which is a consortium partner of the Crescer project).

The standard of care in the MuCCUA trial comprises the recruitment of ADECOS and the community activities they develop, as well as the delivery of preventive pharmacological products through the health posts. The project team selected these activities from those that were described in the National Development Plan of Angola and the Angola Multisectoral Strategic Plan on Nutrition, and are in line with the WHO maternal and infant health care recommendations. Therefore, the research team in this trial is assuring the implementation of the ADECOS figure securing their subsidies and the quality of some other aspects of the standard of care such as providing or securing the stock of deworming,

vitamin A and malaria prophylaxis in the health posts. Trainings and sensitization are done to ensure quality and they are a part of the current ADECOS policy.

The standard of care is provided to all communities participating in the MuCCUA trial. The arm that only receives the standard of care is considered the comparator, as the other two arms include the standard of care plus another intervention, either the unconditional cash transfer or the nutritional supplementation.”

14. Page 11 line 19: Which is approx. how many USD per month? And how does this compare to the minimum wage (or another similar metric) in Angola?

We have added the conversion to USD and information on the Angolan minimum wage to help contextualize this information, page 14. “A total of 13855 kwanzas/month (US\$31.5, Oct22) will be delivered to households with 4 or more inhabitants. A total of 10855 kwanzas/month (US\$24.7, Oct22) will be delivered to households with 3 or fewer inhabitants. The minimum inter professional salary in Angola at the time being is set at 35000 kwanzas per month (US\$79.5, Oct22).”

15. Page 12 lines 12-23: Will cost-effectiveness be evaluated on the basis of each of these secondary outcomes as well as stunting at 24mo? Would be helpful to be explicit about this.

The only outcome that would be included in economic analysis is prevalence of global stunting defined by a z-score <-2. It is written in page 7 under the “Aims and objectives” section and in the first paragraph of the section “Cost-effectiveness analysis” in page 22.

16. Page 12 line 35: I’m not sure any population characteristics were “mentioned earlier”. Not clear what characteristics you are referring to here. We thank the reviewer for this comment and reference. We were actually referring to the heterogeneity/subgroup analysis and we have modified the text accordingly in pages 7 and 13 “heterogeneity analysis will be conducted to assess how costs and impacts of the interventions are equitably distributed among the target population.”

We will perform heterogeneity analysis based on province, commune and access to services, which are mentioned in page 15 lines 37-45. The environment and way of living in each of the two provinces are slightly different in terms of distances between household’s compounds and to health services. For example, in the province of Cunene, households called “kimbos” are very far apart from each other and they rely solely on themselves while in the province of Huila, communities live closer together in villages.

17. Page 12 lines 51-58: Earlier you noted you would measure the costs faced by “programme, provider and households”. Here you describe implementing agencies and users of the interventions. Would be helpful to adopt one set of terms and use them consistently with respect to costs to whom.

We agree with the reviewer and we have modified the text accordingly to be consistent throughout the manuscript. Now in page 15 reads: “The cost-efficiency and cost-effectiveness of the MuCCUA interventions will be measured from a societal perspective (27,28); taking into account costs incurred by the implementing program providers, namely the Institute of Local Development-Social Support Fund of Angola (FAS) and Vall d'Hebron Institute of Research (VHIR) (programme-provider costs), ADECOS (community frontline workers) and by the beneficiaries who are intervention participants, and their households.”

18. Page 15 lines 35-40: Please describe a bit more about what this qualitative data collection will involve and how it will be used to inform cost estimates.

Qualitative data will inform cost estimates on different levels. Key informant interviews with project staff will provide information on time allocation to different activities of the interventions. Focus group discussions and interviews with participants will collect data to inform on opportunity costs of

beneficiaries for participation on the interventions. This information has now been added to the text in page 18.

19. Page 16 lines 46-47: If specific messaging was developed for the “Standard of Care” components of the interventions, it’s not really SOC, right? Anything that is new/different from what normally takes place would be in addition to the SOC.

As explained above the SOC is an intervention designed for the MuCCUA trial based on national recommendations. Please refer to responses to previous comments where it is explained in detail.

20. Page 17 lines 28-31: It’s unclear how information collected from study participants could be used to measure changes in service utilization. Do you mean differences in service utilization between the SOC arms and the other intervention arms? It seems like before/during service utilization records for relevant health facilities would also be important, if such records exist?

The information we collect from participants is related to the use of the health services. It will be measured in the three arms, as they all have SOC, throughout the whole period of the project. Thus, we will be able to compare before and after within each arm, as well as between the arms. Collecting records from health facilities would also be very interesting but out of the scope of our project.

21. Page 17 lines 37-38: How will increased workloads be “costed”? Will they work extra time at their wage rate? Or will existing staff be expected to shoulder an increased workload without compensation? If the latter, even if this is not directly accounted for in the costing, might be important to consider impacts of motivation/mental health/performance.

We agree with the reviewer that assessing the costs of the potential increase in workload for health workers from different perspective would be of great relevance and it could be the subject of a complementary study, but it is out of the scope of this project.

We will assess the increased workload circumscribed to the specific trainings the health technicians receive from the project team in order to provide the medications related to the standard of care (vitamin A, deworming and malaria prophylaxis). These trainings do not involve extra time, as the administration of these medications should be routine of their practice with pregnant women and children. The time and costs of assisting trainings will be calculated with salary tables and time dedicated to them.

22. Page 17 lines 44-48: Again, it’s unclear how new training and sensitization can be considered SOC...

We apologise for the confusion and we provide more background information to help clarify the design and different arms of the trial.

The standard of care is not exactly the status quo in Angola. In 2014, the Ministry of Territorial Development of Angola designed a new community development policy aiming to address human resource shortages in remote areas effectively, in order to provide primary and preventive care, and to improve health status and quality of life in rural communities. This policy is to be implemented by the Angolan Institute of Local Development- Social Support Fund (FAS) and includes the recruitment of two new figures: the sector supervisors of the main extension services (health, social protection, and agriculture), and the ADECOS (Community and Health Development Agents). The ADECOS are trained in the identification of danger signs of malnutrition and referral to a health post and appropriate WASH and dietary practices and work with the families and communities as linkages with the health system (Ministério da Saúde, República de Angola. Política Nacional de Agentes de Desenvolvimento Comunitário e Sanitário (ADECOS), Luanda, Angola 2014)..

In 2020, at the time the MuCCUA trial was designed, the policy was at different stages of implementation across the country, but the Southern provinces had not yet started its implementation at the health care level. Therefore, in the study areas of the provinces of Huila and Cunene, the Crescer project supported the policy by funding the recruitment of ADECOS through the FAS (which is a consortium partner of the Crescer project).

The standard of care in the MuCCUA trial comprises the recruitment of ADECOS and the community activities they develop, as well as the delivery of preventive pharmacological products through the health posts. The project team selected these activities from those that were described in the National Development Plan of Angola and the Angola Multisectoral Strategic Plan on Nutrition, and are in line with the WHO maternal and infant health care recommendations. Therefore, the research team in this trial is assuring the implementation of the ADECOS figure securing their subsidies and the quality of some other aspects of the standard of care such as providing or securing the stock of deworming, vitamin A and malaria prophylaxis in the health posts. Trainings and sensitization are done to ensure quality and they are a part of the current ADECOS policy.

Thus, the standard of care is not exactly the status quo in the whole country, but a set of selected community activities within those recommended in the national plans.

The standard of care is provided to all communities participating in the MuCCUA trial. The arm that only receives the standard of care is considered the comparator, as the other two arms include the standard of care plus another intervention, either the unconditional cash transfer or the nutritional supplementation.

We have amended the text in order to make it more clear (page 5 and page 12).

23. Page 19 lines 11-12: But only if there is a statistically significant difference in these outcomes between SOC and the other intervention arms, right?

We agree with the reviewer, this point was missing and it has been added to the text in page 22.

24. Page 19 lines 10-20: As I mentioned above, I think important to note the limitation that that the economic analysis was conducted in the context of a research project, and identifying and netting out research vs implementation costs can sometimes be difficult.

We agree with the reviewer and we have added this limitation to the text.